# Prognostic Factors in Pseudomyxoma Peritonei with Emphasis on the Predictive Role of Peritoneal Cancer Index and Tumor Markers

**DOI:** 10.3390/cancers15041326

**Published:** 2023-02-19

**Authors:** Sebastian Blaj, David Dora, Zoltan Lohinai, Zoltan Herold, Attila Marcell Szasz, Jonas Herzberg, Roland Kodacsi, Saher Baransi, Hans Jürgen Schlitt, Matthias Hornung, Jens M. Werner, Przemyslaw Slowik, Miklos Acs, Pompiliu Piso

**Affiliations:** 1Department of General and Visceral Surgery, Hospital Barmherzige Brüder, D-93049 Regensburg, Germany; 2Department of Anatomy, Histology and Embryology, Semmelweis University, H-1094 Budapest, Hungary; 3Translational Medicine Institute, Semmelweis University, H-1094 Budapest, Hungary; 4Department of Pulmonology, Pulmonary Hospital Torokbalint, H-2045 Torokbalint, Hungary; 5Division of Oncology, Department of Internal Medicine and Oncology, Semmelweis University, H-1083 Budapest, Hungary; 6Department of Surgery, Krankenhaus Reinbek St. Adolf-Stift, D-21465 Reinbek, Germany; 7Department of Cardiothoracic Surgery, University Medical Center, D-93053 Regensburg, Germany; 8Department of Gynecology and Obstetrics, Florence Nightingale Hospital, D-40489 Düsseldorf, Germany; 9Department of Surgery, University Hospital, 93053 Regensburg, Germany

**Keywords:** pseudomyxoma peritonei, low-grade appendiceal mucinous neoplasms, hyperthermic intraperitoneal chemotherapy, peritoneal cancer index, cytoreductive surgery, biomarkers, tumor, ca-19-9 antigen, carcinoembryonic antigen

## Abstract

**Simple Summary:**

Disease outcome of patients with pseudomyxoma peritonei (PMP) and appendiceal neoplasms improved gradually in the past decades, as their pathology is better understood, together with a refined subtypisation. The introduction of cytoreductive surgery and hyperthermic intraperitoneal chemotherapy in the treatment of well-selected patients has led to significantly better outcomes compared to “classical” therapies alone. Due to the fact that PMP is considered a rare malignancy, real-word cross-sectional studies are highly warranted to establish reliable prognostic and predictive factors in large and uniform patient populations. Apart from established prognostic indicators, such as the peritoneal cancer index, the extent of cytoreduction and the plasma level of tumor markers are as well of significance. We aimed to refine and confirm the validity of these factors in our whole patient population and in patients with low-grade histologies. Our study can refine and improve our current understanding of PMP management on a sizeable cohort.

**Abstract:**

Background: Pseudomyxoma peritonei (PMP) is a rare peritoneal condition where mucus-secreting tumorous cells progressively produce a thick, gelatin-like substance. The prognosis of patients with PMP is determined by the degree of cellularity within the mucin (low-grade (LAMN) vs. high-grade (HAMN) histologic features) and by the extent of the disease. Methods: Prognostic relevance of tumor markers CA19-9 and CEA, gender, Peritoneal Cancer Index (PCI), and completeness of cytoreduction (CC) after cytoreductive surgery were evaluated on 193 consecutive PMP patients, based on a retrospective analysis of prospectively gathered data from a German tertial referral center. Results: We demonstrated that low PCI, CC0 status, low-grade histology, and female gender were independent positive prognostic factors for both overall survival (OS) and progression-free survival (PFS). Furthermore, LAMN patients with achieved CC0 status show significantly better OS and PFS compared to those with CC1 status (*p* = 0.0353 and *p* = 0.0026 respectively). In contrast, the duration and drug of hyperthermic intraperitoneal chemotherapy (HIPEC) were not prognostic in any comparison. Increased CA19-9 and CEA levels were significantly associated with HAMN cases, but also predicted recurrence in patients with low-grade histologies. Conclusion: Our study confirmed the prognostic role of tumor markers and emphasized the importance of CC status and PCI in a large cohort of PMP- and LAMN patients.

## 1. Introduction

Mucinous appendiceal neoplasms are composed of enteric glandular epithelium producing abundant mucin often extending into the appendiceal lumen with extracellular and peritoneal dissemination. The scenario of intraperitoneal accumulation of mucin secondary to mucinous neoplasia leads to the clinical picture of pseudomyxoma peritonei (PMP). The annual incidence of PMP has been estimated at 1–2 per million [1]. Cytoreductive surgery (CRS) with hyperthermic intraperitoneal chemotherapy (HIPEC) has emerged as the treatment of choice in pseudomyxoma peritonei (PMP) [2] and has become the standard treatment for almost all subtypes of mucinous appendiceal neoplasms with peritoneal dissemination [3,4,5]. The classification of mucinous appendiceal neoplasia has been changed several times over the years, which has led to difficulties in comparing and sharing results between centers. The consensus panel of the Peritoneal Surface Oncology Group International (PSOGI) proposed a new classification which has been published in 2016 [6]. Accordingly, there are low-grade appendiceal mucinous neoplasms (LAMN) showing histologically low-grade neoplastic epithelium with effacement or loss of lamina propria [7], and high-grade appendiceal mucinous neoplasms (HAMN). The latter comprises neoplasms with high-grade dysplasia but with no epithelial cancerous invasion [7]. Likewise, the features and terminology of PMP have also been considered in terms of acellular mucin: low-grade and high-grade mucinous carcinoma peritonei as well as high-grade mucinous carcinoma peritonei with signet ring cells, based on the cellularity of the mucin and the degree of atypia of the cellular component [6]. There is a considerable variation in the reported rates of development of PMP from mucinous appendiceal neoplasms, ranging between 20% and 52% [1,8,9].

The carcinoembryonic antigen (CEA) is represented by a group of proteins which are involved in cell adhesion. These proteins are usually produced before birth in the gastrointestinal tract and may appear later on in life as products of certain malignant tumors [10]. Especially colorectal cancer, gastric cancer, pancreatic, and breast cancer are known to express CEA [11,12,13]. In contrast, the carbohydrate antigen 19-9 (CA 19-9) is a saccharin, which is produced especially by pancreatic cancer cells, but also by colorectal- gastric- and bile duct malignancies [14,15,16,17,18]. It acts by ligating E-selectin which is a vascular cell adhesion molecule, facilitating hematogenous metastases [19]. Its main clinical value is in the post-therapeutic assessment of patients with pancreatic cancer. About 10% of Caucasian people who are missing the Lewis blood-type antigen do not produce CA19-9 at all [20].

In the current study, we set the primary goal to demonstrate our experience in the treatment of mucinous appendiceal neoplasms (LAMN and HAMN) dedicating particular interest to recurrence/progression-free (PFS) and overall survival (OS) as well as to other related clinicopathological factors. The present study aims to identify the factors that significantly influence PFS and OS in patients with both LAMN and HAMN based on prospectively gathered data from a German tertial referral center.

## 2. Materials and Methods

### 2.1. Ethical Statement

Due to the pseudonymization of patient data, no institutional ethical approval was needed and the institutional review board (Hospital Barmherzige Brüder Regensburg) approved the study. All the patients included in this study signed an informed consent prior to the surgery and agreed to the use of their medical data for scientific purposes. After data collection, patient-IDs were removed so none of the included individuals can be recognized directly or indirectly.

### 2.2. Study Population

We have retrospectively analyzed prospectively gathered patient data from a German tertial referral center for the therapy of peritoneal surface malignancies with an annual volume of up to 120 procedures. Not only demographical data but also various disease and procedure-related information were included in the HIPEC register of the German Society of General and Visceral surgery.

All patients included in this study (*n* = 193) had been diagnosed with PMP. All patients underwent CRS + HIPEC of their primary (*n* = 168) or recurrent (*n* = 25) tumors. Besides demographical data, also clinical and pathological information were recorded [histology-LAMN or HAMN-, ASA (American Society of Anesthesiologists)-category, and BMI (Body Mass Index)]. We have identified 177 patients with LAMN and 16 with HAMN. For *n* = 142 (80.22%) patients, baseline plasma levels of circulating tumor markers (CEA and CA19-9) were also available. HIPEC was administered using the closed method, and the duration (30, 60, or 90 min) and the drug used (mitomycin C or oxaliplatin) during HIPEC were documented. The peritoneal disease burden was evaluated using the peritoneal cancer index (PCI) according to the Sugarbaker/Jacquet classification [21]. After cytoreduction, the completeness of cytoreduction (CC) score was recorded [21].

### 2.3. Statistical Analysis

Continuous parameters were compared using the Mann–Whitney U-test and Spearman’s correlation coefficient was calculated to evaluate the statistical correlation of clinicopathological parameters. Survival analysis was performed with Kaplan–Meier (KM) curves and a comparison of survival curves with the log-rank test. Cut-offs for KM curves and specificity/sensitivity values for clinical parameters were defined by receiver operating characteristic (ROC) curve analysis using binary outcomes. Cox-proportional hazard regression was used to screen for significant predictor variables. Two-sided analyses with a level of significance of α = 0.05 were performed. The backward elimination method was used for multivariate Cox regression. Only parameters with a significant predictor value (*p* < 0.1) were included in multivariate analyses. To assess the goodness of fit of our multivariate model, Harrel’s C-index (>0.7) was calculated. Statistical analyses were performed using the MedCalc 9.3 software package.

## 3. Results

### 3.1. Cohort Characteristics, Baseline Parameters, and Survival

Our study cohort included 193 patients diagnosed with PMP who underwent CRS + HIPEC. Majority of our patients were diagnosed with LAMN (92.2%), and only 7.8% of the patients were diagnosed with HAMN. 86.8% of the patients underwent surgery of their primary tumor, 13.2% were operated for recurrence. 84.5%, 11.6%, and 3.9% of the patients were assessed with a CC0, a CC1, and a CC2 situation, respectively. The median age at the time of diagnosis was 61 years. Detailed clinicopathological data of the patient cohort are shown in Table 1.

LAMN exhibited significantly increased OS and PFS compared to HAMN both in the whole cohort (*p* < 0.0001, respectively, Figure 1A,B) and in the case of primary tumors. There was no significant difference either in OS or in PFS between patients with primary and recurrent tumors (*p* = 0.8290 and *p* = 0.2840, respectively, Figure 1C,D); however, female LAMN patients exhibited significantly improved outcomes (compared to male patients) concerning both OS (*p* = 0.0337) and PFS (*p* = 0.0059) (Figure 1E,F).

### 3.2. Plasma Concentration of Tumor Markers CA19-9 and CEA Correlates with Histology and PCI Ccore

Plasma levels of key tumor markers CA19-9 and CEA were evaluated before CRS, and patients with HAMN histology showed significantly higher levels of the two tumor markers (Figure 2A,B). CA19-9 and CEA levels were positively correlated with PCI score (Figure 2C,D). Receiver operated characteristic analysis (ROC) showed that both CA19-9 and CEA were fair predictors of disease recurrence (AUC = 0.758 and AUC = 0.723, respectively, Appendix A). Using cut-offs generated by ROC-curve analysis, we compared the survival of high vs. low CA19-9 and high vs. low CEA LAMN patients. Both CA19-9-high and CEA-high patients exhibited significantly decreased OS (*p* = 0.0110, Figure 2E; *p* = 0.0068, Figure 2F) and PFS (*p* = 0.0002, Figure 2G; *p* < 0.0001, Figure 2H). These results underpin the prognostic and predictive value of these tumor markers in PMP and in the case of LAMN patients as well.

### 3.3. CC Score Is Prognostic in LAMN Patients

To assess the prognostic role of CRS and CC-score, we stratified patients according to the completeness of cytoreduction score (CC0, CC1, and CC2 situation). CC score was prognostic when evaluating both OS and PFS in LAMN patients (*p* = 0.0013 and *p* < 0.0001, respectively, Figure 3). Interestingly, when comparing patients with CC0 and CC1 situation, a significant difference was detected regarding both OS (*p* = 0.0353) and PFS (*p* = 0.0026). Patients with HAMN showed no difference in OS according to CC score (*p* = 0.933, Appendix A). In contrast, a modest trend was detected in PFS between patients with CC0 and ≥CC1 (*p* = 0.1580, Figure 2B). Here, the lack of significance might be attributed to low case numbers.

We have also subgrouped patients according to the type of HIPEC medication and duration. The mitomycin C- and oxaliplatin-treated patient groups showed no significant difference regarding PFS either in the whole cohort (*p* = 0.7100, Appendix A), or in LAMN patients (*p* = 0.6390, Appendix A). Concerning HIPEC duration, LAMN patients with 30 or 60 min of HIPEC procedure were compared to patients treated for 90 min with HIPEC and no significant difference was detected in the PFS between the two groups (*p* = 0.7960, Appendix A). For patients with CC0 situation, there was a non-significant trend toward increased PFS in the case of 30–60 min treatments (*p* = 0.0781, Appendix A). There was no significant difference regarding OS in any comparison.

### 3.4. Prognostic Role of PCI Score Status

Next, we explored the prognostic role of peritoneal cancer index (PCI) in PMP. We found that PCI score and OS show no significant correlation (R(s) = −0.042, *p* = 0.544). Nevertheless, there was a weak, but significant negative correlation between PCI score and PFS in the whole cohort (R(s) = −0.237, *p* = 0.001, Figure 4A) and in patients with LAMN (R(s) = −0.228, *p* = 0.0028, Figure 4B). Additionally, ROC analysis showed that PCI score was a fair predictor of recurrence in PMP (AUC = 0.717, Appendix A) and in LAMN patients (AUC = 0.732, Appendix A). After stratifying patients according to PCI score (high vs. low, using the cut-off of PCI = 12), we performed a Kaplan–Meier analysis to assess the role of PCI in PFS. We found that PCI-high LAMN patients exhibited significantly decreased PFS (compared to PCI-low LAMN patients, *p* = 0.0002, Figure 4C). Moreover, performing the analysis for patients with low CC scores (0–1), PCI-high status still conferred significantly decreased PFS to patients (*p* = 0.0003, Figure 4D).

### 3.5. Multivariate Analysis

Cox proportional hazard regression was used to assess the independent prognostic- and predictive roles of studied clinicopathological parameters. We performed univariate Cox regression for all parameters, then, continued with multivariate testing using only significant predictors. Cox-models were performed for OS and PFS, where backwards elimination was used for parameters where, *p* > 0.1000.

According to univariate Cox regression, age (*p* = 0.0284), the plasma level of CA19-9 (*p* = 0.0192) and CEA (*p* = 0.0003), histology (LAMN vs. HAMN, *p* = 0.0002), CC-score (*p* = 0.0011), and PCI score (*p* = 0.0038) were confirmed as independent predictors of OS (Table 2), while gender (*p* = 0.0056), the plasma level of CA19-9 (*p* = 0.0001) and CEA (*p* < 0.0001), histology (LAMN vs. HAMN, *p* < 0.0001), CC-score (*p* = 0.0002), and PCI score (*p* < 0.0001) were confirmed as independent predictors of PFS (Table 3). In the multivariate model, age (*p* = 0.0156), CC-score (*p* = 0.0379), histology (*p* = 0.0091), and PCI score (*p* = 0.0037) remained as independent predictors of OS (Table 4), and PCI score (*p* < 0.0001) and histology (*p* = 0.0112) remained as independent predictors of PFS (Table 5).

## 4. Discussion

It is known that there are several factors which influence the outcome of patients with PMP. These are not only the extent of the disease, but rather the histological subtype (low-grade versus high-grade) and the completeness of surgical cytoreduction. The aim of the present study was to assess the prognostic value of the tumor marker CA19-9 and CEA, as well as to recapitulate the reliability of the PCI on the oncological outcome of these patients.

Several studies have been published in the past decade, which have been dealing with the prognostic value of circulating tumor markers in patients with PMP. Some tumor markers, including CEA, CA19-9, and CA12-5 may be useful in patients who are secretors in order to predict disease aggressiveness, to dictate the intensity of the postoperative scans, and therefore help in early identification of recurrences [22]. A large series of 519 consecutive patients with appendiceal neoplasm was included in a study from a British referral center with the aim to evaluate the value of pre- and postoperative levels of CEA, CA19-9, and CA12-5 and the impact of these values on the prediction of the oncological outcome. OS and disease-free survival (DSF) of patients with normal serum tumor marker levels were significantly better compared to patients with elevated tumor markers. The conclusion of the study included that tumor markers may be useful in the future in predicting the evolution of PMP and other peritoneal malignancies, as they might influence the frequency of postoperative follow-up scans [23]. According to an earlier study, when used for postoperative follow-up, tumor-markers anticipated the radiological appearance of tumor recurrence with up to nine months [24].

A US group studied the significance of serum tumor markers in peritoneal carcinomatosis of appendiceal origin in a population of 282 consecutive patients who underwent CRS and HIPEC. In this group, 176 from 282 patients had at least one tumor marker (CEA, CA19-9, or CA12-5) determined either pre- or postoperatively [25]. Of this group, 81.6% received a complete cytoreduction and 89% of them received HIPEC. The findings of the study suggest that none of the three tumor markers correlated significantly with age, gender, presence of symptoms, presence of extraperitoneal disease, ability to achieve complete cytoreduction, histological subtype, or lymph node positivity [25]. It was also noted that elevated preoperative levels of CEA were not significantly associated with shorter OS and PFS, in contrast with CA19-9 and CA12-5. On the other hand, elevated postoperative serum levels were significantly associated with poorer outcome, but only for CEA [25].

A recent observational study from a Dutch tertial referral center included 225 patients with PMP, who underwent CRS with HIPEC (mitomycin C, 35 mg/m^2^, at 40–41 °C for 90 min or—exceptionally—high dose oxaliplatin at 42–43 °C for 30 min); CEA and CA19-9 were determined routinely in all patients preoperatively, while CA125 was determined inconstantly [26]. The follow-up was performed for ten years or until death and included tumor-marker levels and CT-scans. CEA levels were elevated in 11% of the patients with acellular mucin, 65.8% in patients with disseminated peritoneal adenomucinosis (DPAM), and 63% of the patients with peritoneal mutinous carcinomatosis (PMCA) [26]. Oncological outcomes (OS and PFS) were significantly poorer in patients with DPAM or PMCA, who had 1 or 2 elevated tumor markers preoperatively. Conclusively, serum levels of CEA and CA 19-9 are representing independent prognostic factors together with gender, number of affected regions, and completeness of cytoreduction [26].

Earlier, a German group investigated the predictive value of the PCI on the oncological outcome depending on the histology [27]. The authors showed in a group of 123 patients with peritoneal carcinomatosis that PCI is prognostic in non-mucinous, but not in mucinous peritoneal carcinomatosis and concluded that the multimodal therapy could show its benefits not only in patients with low PCI, but also in those with high-PCI and mucinous peritoneal carcinomatosis [27].

In our study, histology showed a significant association with OS as with PFS, as patients with high-grade PMP had significantly poorer outcomes compared with those with low-grade histology. We showed that preoperative serum levels of CEA and CA19-9 were associated with the oncological outcomes, as patients with elevated tumor markers exhibited decreased OS and PFS. All these results are concordant with the data available in the literature, as pointed out in our multivariate analysis, which showed that age, completeness of cytoreduction, histology, and PCI are independent prognostic factors for OS; on the other hand, PCI and histology represent independent predictors for PFS. We also demonstrated that—by setting a cut-off for PCI at 12 points—the extent of disease measured by PCI in patients with LAMN played a significant role in the PFS. This influence remained, as we restricted the analysis to the patients with macroscopical complete cytoreduction (CC0).

An intriguing result of our study is the influence of gender on the oncological outcome of patients with LAMN. We found that females showed significant better OS (110.47 months vs. 96.08 months) and PFS (99.41 vs. 76.51 months) as compared to males. This aspect has already been described before [22]; however, to date, there is no clear explanation for this phenomenon. Some authors have suggested that females might present earlier in the evolution of the disease due to enlargement of ovarial masses which become symptomatic, speeding up diagnosis [22].

The present analysis also showed that neither the type of drug, nor the duration of hyperthermic perfusion influenced the OS or PFS significantly, when evaluating the whole cohort or LAMN patients only. The heterogeneity of applied drugs and duration of HIPEC originate in the trend change, which occurred after the publication of a multicenter, randomized study with the aim to assess the specific benefit of HIPEC in combination with CRS in patients with peritoneal metastases of colorectal cancer (PRODIGE-7 [28]). The authors concluded that the addition of HIPEC does not lead to an improvement in OS as compared to the group which received only CRS and systemic chemotherapy [28]. Limitations of the PRODIGE-7 study include its heterogeneous patient population and the inclusion of patients with PCI > 15 or previously exposed to oxaliplatin [29]. Nevertheless, after the initial presentation of its results, high-dose oxaliplatin and short HIPEC exposure have been abandoned and most of the centers have applied mitomycin C during HIPEC since [28,30,31].

## 5. Conclusions

The present study demonstrates once again the utility of pre- and postoperative determination of tumor markers, as they can predict the postoperative evolution of patients suffering from PMP, who underwent CRS and HIPEC. Limitation of the present study include its retrospective nature and unicentricity, which are compensated by the large number of included patients. Nevertheless, we managed to point out again the importance of the extent of the disease (measured by PCI), the completeness of cytoreduction (even between CC0 and CC1 status), tumor markers, and gender for the oncological outcome in patients with PMP and LAMN. However, whether solely the earlier diagnosis leads to better outcomes in female patients could not be undoubtedly cleared and needs to be further investigated.

## Figures and Tables

**Figure 1 cancers-15-01326-f001:**
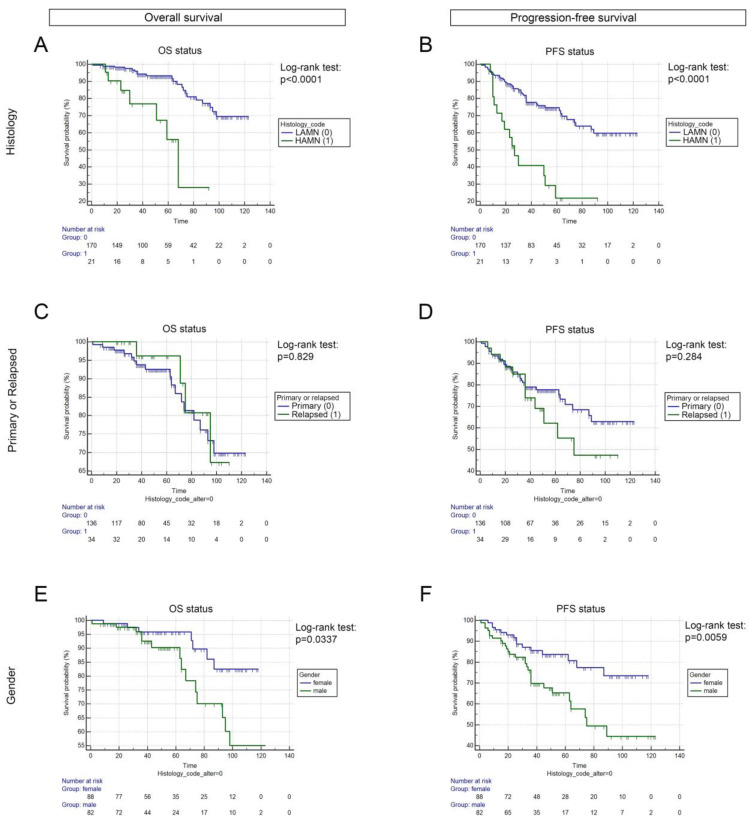
Survival analysis according to histology, tumor type, and gender. According to histology, LAMN patients (vs HAMN patients) showed significantly improved OS (106.45 vs. 61.15 months, *p* < 0.0001, (**A**) and PFS (90.63 vs. 40.52 months, *p* < 0.0001), (**B**)). There was no significant difference neither regarding OS (105.92 vs. 99.42 months, *p* = 0.8290, (**C**) nor PFS (93.23 vs. 74.22 months, *p* = 0.2840), (**D**)) between patients with primary vs. relapsed LAMN tumors. Female LAMN patients exhibited significantly improved OS and PFS compared to male patients (OS: 110.47 vs. 96.08 months, *p* = 0.0337, (**E**); PFS: 99.41 vs. 76.51 months, *p* = 0.0059, (**F**)). Months for OS and PFS are described as means.

**Figure 2 cancers-15-01326-f002:**
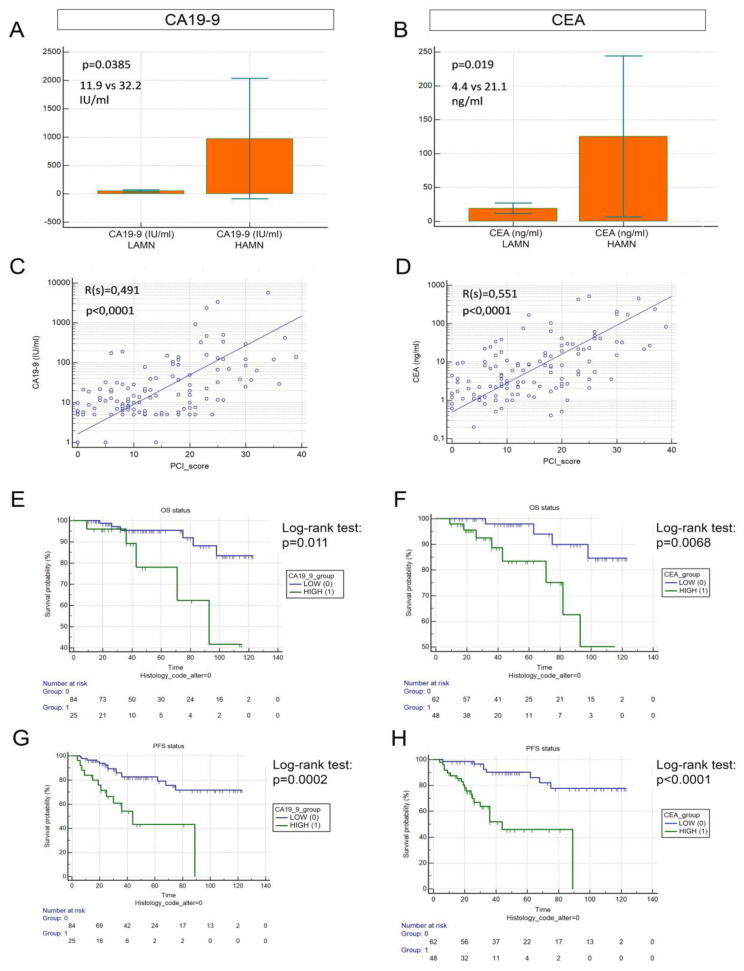
Tumor markers in PMP. CA19-9 and CEA plasma levels were significantly increased in HAMN patients compared to LAMN patients (**A**,**B**). PCI showed a moderate positive correlation with the plasma levels of both tumor markers according to Spearman’s rank correlation (**C**,**D**). Both CA19-9-low and CEA-low patients exhibited significantly better outcomes (compared to CA19-9- and CEA-high patients) including OS (CA19-9: 114.16 vs. 85.88 months, *p* = 0.0110 (**E**); CEA: 115.47 vs. 90.43 months, *p* = 0.0068 (**F**)) and PFS (CA19-9: 99.7 vs. 51.93 months, *p* = 0.0002 (**G**); CEA: 106.7 vs. 54 months, *p* < 0.0001 (**H**)). Months for OS and PFS are described as means.

**Figure 3 cancers-15-01326-f003:**
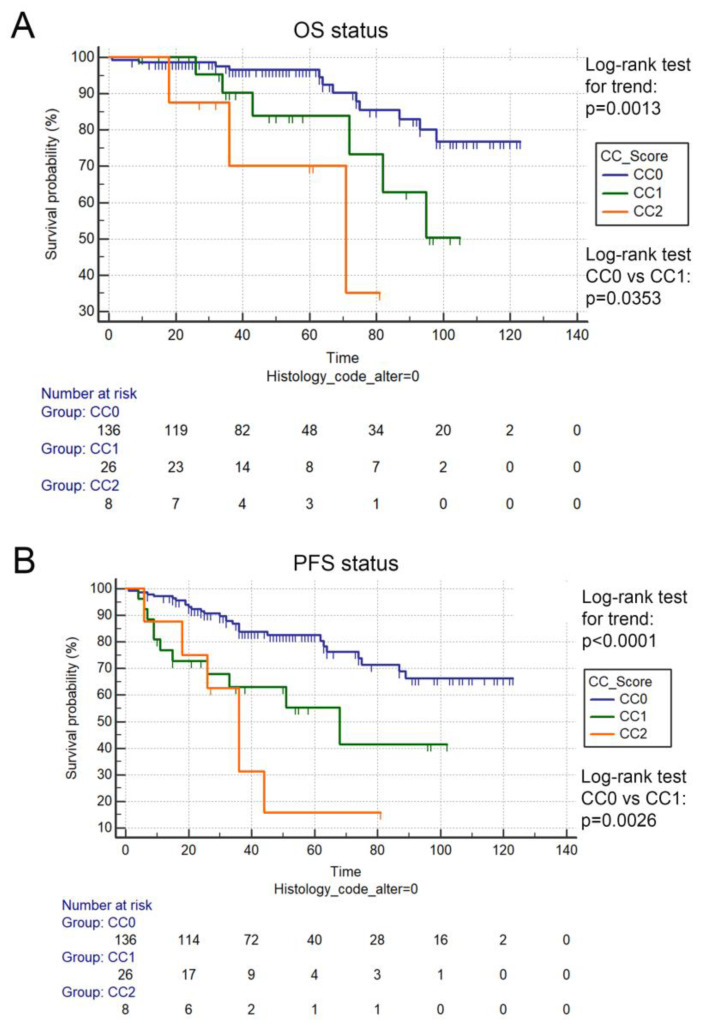
Prognostic role of CC score in LAMN. CC score was prognostic when using Log-rank test for trend, when comparing LAMN patients with CC0, CC1, and CC2 scores in the case of OS (*p* = 0.0013, (**A**) and PFS (*p* < 0.0001), (**B**)). Interestingly, when comparing LAMN patients with CC0 vs. CC1 situation, there was a significant difference in OS (110.8 vs. 86.5 months, *p* = 0.0353, (**A**) and PFS (97.8 vs. 60.8 months, *p* = 0.0026), (**B**)). Months for OS and PFS are described as means.

**Figure 4 cancers-15-01326-f004:**
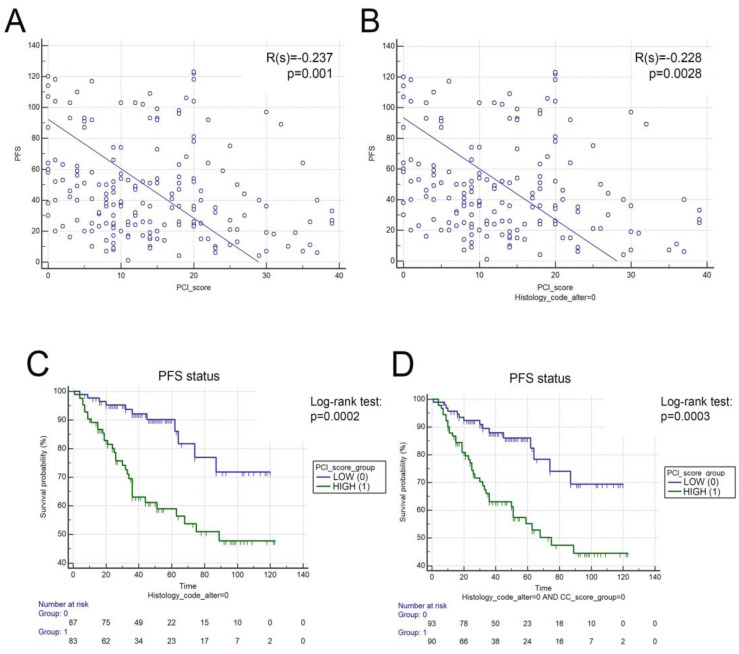
Predictive role of PCI in LAMN. PCI showed a significant, but weak negative correlation with PFS in the whole cohort and in LAMN patients. Low-PCI patients exhibited significantly increased PFS (compared to high-PCI patients), when evaluating all LAMN patients (101.95 vs. 76.88 months, *p* = 0.0002, E) and only LAMN patients with CC0 situation (98.51 vs. 74.06 months, *p* = 0.0003, F). Months for OS and PFS are described as means.

**Table 1 cancers-15-01326-t001:** Clinicopathological characteristics of study cohort. Cut-offs for CA19-9 and CEA plasma levels and for PCI were defined by receiver operating characteristic (ROC) curve analysis using binary outcomes.

*Parameter*	*N*	*Percent*
Gender Male Female	100 93	51.9% 48.1%
Age <60 ≥60	89 104	46.2% 53.8%
Tumor type at CRS primary recurrence	168 25	86.8% 13.2%
Disease recurrence no yes	150 43	77.8% 22.2%
Tumor histology LAMN HAMN	178 15	92.2% 7.8%
ASA status ASA I ASA II ASA III	4 33 156	2.1% 17.1% 80.8%
BMI * <30 kg/m^2^ ≥30 kg/m^2^	105 22	54.4% 11.3%
CA19-9 * <40 IU/mL ≥40 IU/ml	91 32	47.1% 16.5%
CEA * <3 ng/mL ≥3 ng/ml	66 58	34.1% 30%
HIPEC Duration 30 min 60 min 90 min	124 20 49	64.3% 10.4% 25.3%
HIPEC drug * Mytomycin C Oxaliplatin	67 122	34.7% 63.3%
CC score CC0 CC1 CC2	163 22 8	84.5% 11.6% 3.9%
PCI ≤12 >12	91 102	47.2% 52.8%

* Missing values: BMI (34.3%), CA19-9 (36.4%), CEA (35.9%), HIPEC drug (2%).

**Table 2 cancers-15-01326-t002:** Univariate Cox regression for OS. HR: hazard rate, CI: confidence interval.

Covariate	*p*-Value	HR	95% CI of HR
Gender	0.0875	1.9429	0.9105–4.1461
Age	**0.0284**	1.0337	1.0037–1.0647
ASA status	0.9988	0.9995	0.5360–1.8639
BMI (kg/m^2^)	0.9077	1.0078	0.8845–1.1483
CA19-9 IU/mL	**0.0192**	1.0006	1.0001–1.0011
CEA ng/mL	**0.0003**	1.0086	1.0040–1.0133
Histology (LAMN vs. HAMN)	**0.0002**	5.6928	2.3236–13.947
Primary/Recurrence	0.8298	1.0986	0.4680–2.5789
HIPEC Duration	0.3905	0.4012	0.0504–3.1916
HIPEC Drug	0.3529	0.4929	0.1116–2.1761
CC score	**0.0011**	2.3344	1.4079–3.8708
PCI	**0.0038**	1.0575	1.0183–1.0981

Bold: the significant values.

**Table 3 cancers-15-01326-t003:** Univariate Cox regression for PFS. HR: hazard rate, CI: confidence interval.

Covariate	*p*-Value	HR	95% CI of HR
Gender	**0.0056**	2.1289	1.2506–3.6240
Age	0.9672	1.0004	0.9823–1.0188
ASA status	0.9408	1.0170	0.6541–1.5811
BMI (kg/m^2^)	0.1831	0.9448	0.8694–1.0267
CA19-9 IU/mL	**0.0001**	1.0005	1.0003–1.0008
CEA ng/mL	**<0.0001**	1.0063	1.0038–1.0088
Histology (LAMN vs. HAMN)	**<0.0001**	3.8875	2.1563–7.0085
Primary/Recurrence	0.3154	1.3421	0.7579–2.3767
HIPEC Duration	0.0705	1.8378	0.9535–3.5423
HIPEC Drug	0.4260	1.2857	0.6947–2.3794
CC score	**0.0002**	1.9795	1.3922–2.8144
PCI	**<0.0001**	1.0719	1.0455–1.0990

Bold: the significant values.

**Table 4 cancers-15-01326-t004:** Multivariate Cox regression for OS. HR: hazard rate, CI: confidence interval.

Covariate	*p*-Value	HR	95% CI of HR
Age	**0.0156**	1.0519	1.0099–1.0958
CC score	**0.0379**	2.1954	1.0490–4.5947
Histology (LAMN vs. HAMN)	**0.0091**	6.6807	1.6139–27.6553
PCI	**0.0037**	1.1006	1.0319–1.1740

Bold: the significant values.

**Table 5 cancers-15-01326-t005:** Multivariate Cox regression for PFS. HR: hazard rate, CI: confidence interval.

Covariate	*p*-Value	HR	95% CI of HR
PCI	**<0.0001**	1.1122	1.0707–1.1553
Histology (LAMN vs. HAMN)	**0.0112**	2.6247	1.2503–5.5100

Bold: the significant values.

## Data Availability

The data presented in this study are available on request from the corresponding author.

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
