# Peer review of "Prognostic Factors in Pseudomyxoma Peritonei with Emphasis on the Predictive Role of Peritoneal Cancer Index and Tumor Markers"

_cancers, 2023, doi:10.3390/cancers15041326_

Round 1
Reviewer 1 Report
The introduction is well written and is sufficient to guide the reader towards the manuscript. First of all the authors should be congratulated for prospectively collecting all these data of a rare tumor. Including 193 patients for such a rare tumor is great!
Table 1 is quite confusing. There is nog legend for group A vs group B (I suppose low grade vs high grade? Also choosing the opposite directing for each group of presenting the results is very confusing.
The results are well presented and the discussion underpins the results and nicely reviews the literature.
Consider revising table 1 to make this more easy to read.
Overall only minor comments, but the authors can be congratulated for this retrospective and monocentric study, but with a high number of inclusions for a rare tumor
Author Response
We thank the expert reviewers for pointing this out. Accordingly, we changed the format of Table 1 to be easier to read and omitted the group labeling. Since our study did not analyze outcomes focusing on a singular, but on multiple binary- and continuous parameters with equal emphasis, we could not use a two-column format for two separate patient groups. Thus, we simplified our table format and showed different patient groups in the whole cohort.
Reviewer 2 Report
Blaj et al. demonstrated the utility of pre- and post-operative determination of tumor markers and importance of PCI, cytoreduction, tumor markers, and gender for the oncological outcome in PMP and LAMN patients. The data was clearly presented with appropriate analysis. The manuscript was well-written.
1. Table 1. Groups A and B were grouped based on histology. In Group A, was 51.9% male based on total 193 patients or 51.9% of 177 patients in group A? The table was not clear.
2 2. Baseline plasma levels of CEA and CA19-9 before CRS showed positive correlation with PCI score and value in PMP and LAMN patients. Authors discussed the predictive and prognostic value of these markers pre and post CRS. Can authors discuss whether there is any difference on predictive value on disease recurrence or OS using pre or post CRS plasma?
Author Response
We thank the expert reviewers for pointing this out. Accordingly, we changed the format of Table 1 to be easier to read and omitted the group labeling.
Unfortunately due to inconstant determination (mainly external follow-up) of postoperative values of the tumor markers we were not able to determine their role in the oncological outcome of these patients.